# Parameterization of the Stochastic Model for Evaluating Variable Small Data in the Shannon Entropy Basis

**DOI:** 10.3390/e25020184

**Published:** 2023-01-17

**Authors:** Oleh Bisikalo, Vyacheslav Kharchenko, Viacheslav Kovtun, Iurii Krak, Sergii Pavlov

**Affiliations:** 1Department of Automation and Intelligent Information Technologies, Faculty of Intelligent Information Technologies and Automation, Vinnytsia National Technical University, Khmelnitske Shose Str. 95, 21000 Vinnytsia, Ukraine; 2Department of Computer Systems, Networks and Cybersecurity, Faculty of Radio Electronics, Computer Systems and Information Communications, National Aerospace University KhAI, Chkalov Str. 17, 610700 Kharkiv, Ukraine; 3Department of Computer Control Systems, Faculty of Intelligent Information Technologies and Automation, Vinnytsia National Technical University, Khmelnitske Shose Str. 95, 21000 Vinnytsia, Ukraine; 4Department of Theoretical Cybernetics, Faculty of Computer Sciences and Cybernetics, Taras Shevchenko National University of Kyiv, Volodymyrska Str. 60, 01033 Kyiv, Ukraine; 5Laboratory of Biomedical Optics, Department of Biomedical Engineering, Faculty for Infocommunications, Radioelectronics and Nanosystems, Vinnytsia National Technical University, Khmelnitske Shose Str. 95, 21000 Vinnytsia, Ukraine

**Keywords:** Shannon entropy, machine learning, evaluation of small data, measurement errors, stochastic model, parametric optimization, normalized probabilities, interval probabilities

## Abstract

The article analytically summarizes the idea of applying Shannon’s principle of entropy maximization to sets that represent the results of observations of the “input” and “output” entities of the stochastic model for evaluating variable small data. To formalize this idea, a sequential transition from the likelihood function to the likelihood functional and the Shannon entropy functional is analytically described. Shannon’s entropy characterizes the uncertainty caused not only by the probabilistic nature of the parameters of the stochastic data evaluation model but also by interferences that distort the results of the measurements of the values of these parameters. Accordingly, based on the Shannon entropy, it is possible to determine the best estimates of the values of these parameters for maximally uncertain (per entropy unit) distortions that cause measurement variability. This postulate is organically transferred to the statement that the estimates of the density of the probability distribution of the parameters of the stochastic model of small data obtained as a result of Shannon entropy maximization will also take into account the fact of the variability of the process of their measurements. In the article, this principle is developed into the information technology of the parametric and non-parametric evaluation on the basis of Shannon entropy of small data measured under the influence of interferences. The article analytically formalizes three key elements: -instances of the class of parameterized stochastic models for evaluating variable small data; -methods of estimating the probability density function of their parameters, represented by normalized or interval probabilities; -approaches to generating an ensemble of random vectors of initial parameters.

## 1. Introduction

One of the most relevant problems of modern science is the extraction of useful information from available data. In various fields of science, methodologies aimed at solving this problem are being developed. Each such methodology is based on a certain hypothesis about the properties of the data and the real or hypothetical source of their origin. In the context of the data evaluation problem, two fundamental hypotheses can be distinguished [1,2,3,4,5]. The first hypothesis focuses on directly measurable, deterministic parameters to identify potential functional dependencies between them. All data that cannot be attributed to one or more defined parameters are considered influences in this hypothesis and are rejected. Naturally, such an approach is adequate and productive only if the information is extracted from data obtained from a known, sufficiently investigated the source of origin. The second hypothesis focuses on the analysis of the data as such and is focused on identifying patterns in them, the presence of which can be assessed using a certain defined metric. This can be, for example, a measure of data sufficiency, a property of a sample from the general population, the normality of probability distribution densities, etc. It is practically impossible to guarantee the characteristics of these properties for specific data. However, the improbable becomes common if we analyze not data, but Big Data. This trend is the basis for the progress of such methodologies as mathematical statistics [2,6,7,8], machine learning [9,10,11,12], econometrics [13,14,15,16], financial mathematics [17,18,19] and control theory [20,21,22,23].

In recent decades, the first two of the methodologies just mentioned have been heard. Machine learning is based on the axiomatic perception of probability spaces, as outlined in the paradigm of the theory of statistical learning developed in the 1960s [24,25,26]. There are several dominant categories of machine learning, but the most common is tutored learning [9,10,27,28]. In this category, researchers work with symmetric finite datasets, summarized in the “input” and “output” entities. The purpose of data analysis is to identify the functional dependence between these entities. The set of admissible types of functions forms the hypothesis space of this category of machine learning. The machine learning algorithm consistently evaluates the expected risks of describing the dependence of the existing “input” and “output” entities by each type of function from the hypothesis space. The evaluation is carried out by calculating a single loss function for the entire research. The expected risk is understood as the product of the sum of the estimates and the probability distribution of the data. If the compatible mapping probability distribution is known, then finding the best hypothesis is a trivial task. In the general case, the distribution is unknown, so the machine learning algorithm chooses the most appropriate hypothesis according to a certain rule and proves this thesis by calculating the empirical risk. In addition to the computational complexity, the disadvantage of machine learning is the tendency of the algorithms of this methodology to minimize the loss function by overfitting the potentially best hypothesis to the available data (so-called overtraining [9,27,29]). A typical way to detect (but not prevent) overtraining is to test the best hypothesis on data that the algorithm has not yet worked on (the control sample). Methods of mathematical statistics are not subject to retraining, because they do not assess empirical risk as such.

A typical example of a problem, in the process of solving which the characteristic features of mathematical statistics and machine learning are manifested, is linear regression [7,8,9,10,11]. In the classic formulation of this problem, we need to find the regression coefficients that minimize the root mean square error between the reference entity “output” and its pattern as generated by the model. Such a problem can be solved in a closed form. The theory of statistical learning states that, if we choose the root mean square error as the loss function and carry out empirical risk optimization, then the obtained result will coincide with the one that we will obtain by applying traditional linear regression analysis. However, the maximum likelihood method [2,6,7,30] characteristic for mathematical statistics will demonstrate a similar result in this situation. By the way, the methods of mathematical statistics do not operate with the concepts of initial and test samples, but use metrics to evaluate the results of the model. In our example, the statistical approach allows us to reach the optimal solution because the solution itself exists in a closed form. The maximum likelihood method does not test alternative hypotheses and does not converge to the optimal solution, unlike a machine learning algorithm. However, if the piecewise linear loss function is used for the machine learning algorithm in the same problem, the final result does not coincide with the maximum likelihood method. The machine learning algorithm allows us to expand the space of relevant hypotheses with an a priori considered loss function. The process of their evaluation is carried out automatically. The maximum likelihood method can estimate the accuracy of the original model but does not allow us to automatically change its appearance. Therefore, the methods of machine learning and mathematical statistics work in different ways, while producing similar results. If the task of the researcher is to accurately predict the cost of housing, then machine learning tools are exactly what is needed. If a scientist is investigating the relationships between parameters or making scientifically based conclusions about the data, then a statistical model cannot be dispensed with.

Finally, machine learning experts say, “There are no such things as unsolvable problems, either data or computing power is scarce”. Indeed, everyone has heard about Big Data analysis [10,11,12,31]. Now, however, the issue of analyzing so-called “small data” is becoming increasingly common [32,33]. Classical machine learning approaches are helpless in such a situation. This circumstance prompted the authors to write this article.

Taking into account the strengths and weaknesses of the mentioned methods, we will formulate the necessary attributes of scientific research.

The **object of the research** is the process of the parameterization of the stochastic model for evaluating variable small data for machine learning purposes.

The **research subject** is probability theory and mathematical statistics, evaluation theory, information theory, mathematical programming methods and experiment planning theory.

The **research aims** to formalize the process of finding the best estimates of the probability density functions for the characteristic parameters of instances of the class of stochastic models for evaluating variable small data.

**The research objectives** are:

(1) To formalize the process of calculating the variable entropy estimation of the probability density functions of the characteristic parameters of the stochastic variable small data estimation model, represented by normalized probabilities;

(2) To formalize the process of calculating the variable entropy estimation of the probability density functions of the characteristic parameters of the stochastic variable small data estimation model, represented by interval probabilities;

(3) To justify the adequacy of the proposed mathematical apparatus and demonstrate its functionality with an example.

The **main contribution** of the research is that the article analytically summarizes the idea of applying the Shannon entropy maximization principle to sets that represent the results of observations of the “input” and “output” entities of the stochastic model for evaluating variable small data. To formalize this idea, a sequential transition from the likelihood function to the likelihood functional and the Shannon entropy functional is analytically described. Shannon’s entropy characterizes the uncertainty caused not only by the probabilistic nature of the parameters of the stochastic data evaluation model but also by influences that distort the results of the measurements of the values of these parameters. Accordingly, based on the Shannon entropy, it is possible to determine the best estimates of the values of these parameters for maximally uncertain (per entropy unit) influences that cause measurement variability. This postulate is organically transferred to the statement that the estimates of the probability distribution density of the parameters of the stochastic model of small data obtained as a result of Shannon entropy maximization will also take into account the fact of the variability of the process of their measurements. In the article, this principle is developed into the information technology of parametric and non-parametric evaluation on the basis of Shannon entropy of small data measured under the influence of interferences.

The **highlights** of the research are:

(1) Instances of the class of parameterized stochastic models for evaluating variable small data;

(2) Methods of estimating the probability density function of their parameters, represented by normalized or interval probabilities;

(3) Approaches to generating an ensemble of random vectors of initial parameters;

(4) A technique for statistical processing of such an ensemble using the Monte Carlo method to bring it to the desired numerical characteristics.

## 2. Models and Methods

### 2.1. Statement of the Research

Evaluation based on data that represent parametric signals or phenomena of physical, medical, economic, biological and other sources of origin is the functional purpose of evaluation theory as a branch of mathematical statistics. To solve the problem of evaluation, parametric and non-parametric approaches are used. In recent decades, the latter has noticeably dominated the former, which has become possible thanks to the “reactive” progress in the field of machine learning and artificial intelligence. At the same time, the focus of researchers’ interest is shifting from the study of the processes represented by Big Data to that of those processes about which the amount of data small, and the data itself contains errors. Such a preamble encourages the perception of the parameters of the small data evaluation model as stochastic quantities. Accordingly, we will call such a model a stochastic model for small data evaluation. The characteristics of such a model are the probability density functions of the stochastic parameters. The primary task in identifying a stochastic estimation model for specific small data is to estimate the parameters of these probability density functions. If this step is passed, then the identified stochastic evaluation model can be taken as a basis for forming moment models of small data, generating an ensemble of random vectors of the initial parameters and carrying out the statistical processing of such an ensemble using the Monte Carlo method [6,7,8] to bring it to the desired numerical characteristics. The formalization of the way to solve the primary problem formulated above has scientific potential and applied value.

Let there be a stochastic parameterized research object represented by the results of measurements, in which the matrix of values of the input parameters X with the dimension [o×n] (entity “input”) is matched by a vector of values of the output parameter y with the dimension [o×1] (entity “output”), where o is the number of censored observations, and n is the number of input characteristic parameters of the research object.

The process of measuring the values of matrix X and vector y is characterized by errors, which are represented by the symmetrical matrix $N=\left(\nu_{ji}\right)$ (variability of the measurement process), $i=\bar{1,n}$, $j=\bar{1,o}$, and vector $\upsilon =\left(\upsilon_{i}\right)$, where $\nu_{ji}$, $\upsilon_{i}$ are independent stochastic values, ∀i,j. The value of these stochastic quantities belongs to the intervals $N_{ji}=\left[\nu_{ji}^{-},\nu_{ji}^{+}\right]$ and ${ϒ}_{j}=\left[\upsilon_{j}^{-},\upsilon_{j}^{+}\right]$, respectively.

The stochastic model of the 〈X,y〉 data evaluation is represented by an expression
(1)s=F(X+N,α)+υ,
where F is a defined o-dimensional vector function, α is a random n-dimensional vector formed by independent stochastic parameters $\alpha_{i}$, $i=\bar{1,n}$, $\forall \alpha_{i}\in A_{i}=\left[\alpha_{i}^{-},\alpha_{i}^{+}\right]$.

Let us assume that the parameters of the stochastic model and the variability of the measurements are continuous stochastic quantities, the values of which belong to the corresponding intervals of the tuple $\langle N_{ji},{ϒ}_{j},A_{i}\rangle$ (hereinafter—the “genuine” version of the stochastic Model (1) or GνSM).

In this case, the probability density functions of the stochastic parameters of GνSM (variability of measurements P(α), input W(N) and output Q(υ) parameters) (Independent Stochastic Parameters of the Small Data Estimation Model) are described by the expressions:(2)$$P\left(\alpha \right)=\prod_{i=1}^{n}p_{i}\left(\alpha_{i}\right),$$
(3)$$W\left(N\right)=\prod_{j=1}^{o}\prod_{i=1}^{n}w_{ji}\left(\nu_{ji}\right),$$
(4)$$Q\left(\upsilon \right)=\prod_{j=1}^{o}q_{j}\left(\upsilon_{j}\right),$$
where $\alpha_{i}\in A_{i}$, $\nu_{ji}\in N_{ji}$ and $\upsilon_{j}\in {ϒ}_{j}$, respectively. Formulating Expressions (2)–(4), the authors implied a priori that the measurement results were obtained in accordance with the provisions of the experiment planning theory. The corresponding variables are statistically independent.

Functions (2)–(4) will be evaluated based on data 〈X,y〉 according to Model (1), taking into account the available a priori information summarized by the tuple $\langle P^{0}\left(\alpha \right),W^{0}\left(N\right),Q^{0}\left(\upsilon \right)\rangle$.

The stochastic Model (1) generates an ensemble of random vectors s, which can be compared with the vector υ obtained as a result of measurements. To carry out such an estimation of the probability density Functions (2)–(4), we will use k moments of the stochastic components of the vector s:$$m^{\left(k\right)}=\left\{M\left(s_{j}^{\left(k\right)}\right)\right\},\;j=\bar{1,o},$$
where (Numerical characteristics for estimating these stochastic parameters)
$$M\left(s_{j}^{\left(k\right)}\right)=\int_{\begin{array}{l} \alpha \in A,\\ \nu \in N,\\ \upsilon \in ϒ \end{array}}{\left(F_{j}\left(X+N,\alpha \right)+\upsilon_{j}\right)}^{k}dP\left(\alpha \right)dW\left(N\right)dQ\left(\upsilon \right).$$

Next, we will use moments of the first order (k=1). In accordance:(5)$$M\left(s\right)=\bar{s}=\int_{\begin{array}{l} \alpha \in A,\\ \nu \in N,\\ \upsilon \in ϒ \end{array}}(F\left(X+N,\alpha \right)+\upsilon )dP\left(\alpha \right)dW\left(N\right)dQ\left(\upsilon \right).$$

Another version of the implementation of the Model (1) will be one in which the parameters of the stochastic model and the variability of the measurements are continuous stochastic values, the belonging of which to the corresponding interval of the tuple $\langle N_{ji},{ϒ}_{j},A_{i}\rangle$ will be characterized by a certain probability (hereinafter—the “quasi” version of the stochastic Model (1) or QνSM). In this case:

(1) the parameters $\alpha_{i}$ take values in $A_{i}$ intervals with probabilities $p_{i}\in \left[0,1\right]$, $i=\bar{1,n}$;

(2) the parameters $\nu_{ji}$ take values in intervals $N_{ji}$ with probabilities $w_{ji}\in \left[0,1\right]$, $j=\bar{1,o}$, $i=\bar{1,n}$;

(3) the parameters $\upsilon_{j}$ take values in intervals ${ϒ}_{j}$ with probabilities $q_{j}\in \left[0,1\right]$, $j=\bar{1,o}$.

The available a priori information is summarized by the vector $\left(p_{i}^{0},w_{ji}^{0},q_{j}^{0}\right)$, $j=\bar{1,o}$, $i=\bar{1,n}$.

At the same time, Expressions (2)–(4) retain their legitimacy. We generalize the initial numerical characteristics of QνSM in the form of a vector of quasi-momentums of the first order:(6)$$\alpha =\alpha^{-}+PL_{\alpha },\;\nu =\nu^{-}+W⊗L_{\nu },\;\upsilon =\upsilon^{-}+QL_{\upsilon },$$
where $L_{\alpha }=diag\left(\alpha_{i}^{+}-\alpha_{i}^{-}|i=\bar{1,n}\right)$, $L_{\nu }=diag\left(\nu_{ji}^{+}-\nu_{ji}^{-}|j=\bar{1,o},i=\bar{1,n}\right)$, $L_{\alpha }=diag\left(\upsilon_{j}^{+}-\upsilon_{j}^{-}|j=\bar{1,o}\right)$ and the ⊗ sign represents the element-by-element multiplication operation. Expressions (6) declare the replacement of the elements of the tuple 〈α,ν,υ〉 with their quasi-average values.

The analytical expression for the first-order quasi-momentum of the stochastic vector s can be obtained by substituting numerical Characteristics (6) into Expression (1):(7)$$\tilde{s}=F\left(\nu^{-}+X+W⊗L_{\nu },\alpha^{-}+PL_{\alpha }\right)+\upsilon^{-}+QL_{\upsilon }.$$

In the context of the proposed statement of the research, we specify its aim and objectivities.

The research aims to formalize the process of finding the best estimates of the probability density functions for the 〈p,q〉 parameters of GνSM and QνSM represented by Expressions (5) and (7), respectively.

The objectives of the research are:

(1) To formalize the process of calculating the variable entropy estimation of the probability density functions of characteristic parameters of GνSM represented by normalized probabilities;

(2) To formalize the process of calculating the variable entropy estimation of the probability density functions of characteristic parameters of QνSM represented by interval probabilities;

(3) To justify the adequacy of the proposed mathematical apparatus and demonstrate its functionality with an example.

### 2.2. Parameterization of the Stochastic Model for Evaluating Variable Small Data in the Shannon Entropy Basis

Let us formulate the corresponding probability functionals for the available information about the values of the input and output parameters of the stochastic Model (1).

Taking into account the independence of the parameters of the “input” and “output” entities in the stochastic Model (1) and the variability of their measurement procedure, we determine the compatible probability density function Φ(α,ν,υ) and the corresponding logarithmic likelihood ratio ϕ(α,ν,υ) as
(8)Φ(α,ν,υ)=P(α)W(ν)Q(υ),
(9)$$\phi \left(\alpha ,\nu ,\upsilon \right)=\ln\frac{P\left(\alpha \right)}{P^{0}\left(\alpha \right)}+\ln\frac{W\left(\nu \right)}{W^{0}\left(\nu \right)}+\ln\frac{Q\left(\upsilon \right)}{Q^{0}\left(\upsilon \right)}.$$

Based on Expressions (8) and (9), we formulate the likelihood functional L(P(α),W(ν),Q(υ)):(10)$$\begin{array}{c} L\left(P\left(\alpha \right),W\left(\nu \right),Q\left(\upsilon \right)\right)=\int_{\begin{array}{l} \alpha \in A,\\ \nu \in N,\\ \upsilon \in ϒ \end{array}}\Phi \left(\alpha ,\nu ,\upsilon \right)\phi \left(\alpha ,\nu ,\upsilon \right)d\alpha d\nu d\upsilon =\int_{\alpha \in A}P\left(\alpha \right)\ln\frac{P\left(\alpha \right)}{P^{0}\left(\alpha \right)}d\alpha \\ +\int_{\nu \in N}W\left(\nu \right)\ln\frac{W\left(\nu \right)}{W^{0}\left(\nu \right)}d\nu +\int_{\upsilon \in ϒ}Q\left(\upsilon \right)\ln\frac{Q\left(\upsilon \right)}{Q^{0}\left(\upsilon \right)}d\upsilon . \end{array}$$

Expression (10) presented in the −L(P(α),W(ν),Q(υ)) format is the Shannon entropy functional [34,35]. According to its purpose, such a functional is a measure for evaluating the degree of variability of the elements of a tuple 〈P(α),W(ν),Q(υ)〉. This fact determines the perspective of using such a functional for evaluating Functions (2)–(4). In the context of this motivation, let us transform Expression (10) into the form
(11)$$\begin{array}{c} H\left(\bar{s}\right)=-\sum_{i=1}^{n}\int_{\alpha_{i}\in A_{i}}p_{i}\left(\alpha_{i}\right)\ln\frac{p_{i}\left(\alpha_{i}\right)}{p_{i}^{0}\left(\alpha_{i}\right)}d\alpha_{i}-\sum_{j=1}^{o}\sum_{i=1}^{n}\int_{\nu_{ji}\in N_{ji}}w_{ji}\left(\nu_{ji}\right)\ln\frac{w_{ji}\left(\nu_{ji}\right)}{w_{ji}^{0}\left(\nu_{ji}\right)}d\nu_{ji}\\ -\sum_{j=1}^{o}\int_{\upsilon_{j}\in {ϒ}_{j}}q_{j}\left(\upsilon_{j}\right)\ln\frac{q_{j}\left(\upsilon_{j}\right)}{q_{j}^{0}\left(\upsilon_{j}\right)}d\upsilon_{j}. \end{array}$$

The Functional (11) is defined for estimating the probability density functions of stochastic parameters of GνSM. For QνSM, based on Expression (10), we obtain:$$H\left(\tilde{s}\right)=-\sum_{i=1}^{n}p_{i}\ln\frac{p_{i}}{p_{i}^{0}}-\sum_{j=1}^{o}\sum_{i=1}^{n}w_{ji}\ln\frac{w_{ji}}{w_{ji}^{0}}-\sum_{j=1}^{o}q_{j}\ln\frac{q_{j}}{q_{j}^{0}}.$$

Based on Definition (11), we formulate the problem of finding the optimal estimate of the probability density functions of stochastic parameters of GνSM, taking into account the fact of their variability, i.e., $E_{\bar{s}}$.

We define the objective function of such an optimization problem as:(12)$$H\left(\bar{s}\right)\to \max.$$

We define the restrictions of the $E_{\bar{s}}$ optimization problem as
(13)E=P∪W∪Q,
that is, the probability distribution density of the variability of measurements P(α)∈P, input W(N)∈W and output Q(υ)∈Q parameters of GνSM must belong to the space E defined by Expression (13), and
(14)$$y=\frac{1}{{\left(M\left(F^{\left(k\right)}\left(X+\nu ,a\right)+\upsilon^{\left(k\right)}\right)\right)}^{k}},$$
that is, the elements of the vector with the results of measurements y are equal to the elements of the kth moment of the vector s raised to the $k^{-1}$th power.

By analogy with the formulation of the optimization Problem (12)–(14), we formulate the problem of finding the optimal estimate of the probability density functions of stochastic parameters of QνSM, taking into account the fact of their variability, i.e., $E_{\tilde{s}}$.

We define the objective function of such an optimization problem as:(15)$$H\left(\tilde{s}\right)\to \max.$$

Recall that the complex parameter $\bar{s}$ generalizes a tuple of interval controlled parameters 〈P(α),W(ν),Q(υ)〉 (see Expressions (10) and (5)), and the complex parameter $\tilde{s}$ focuses on the variability of measuring these characteristic parameters (see Expression (7)).

Considerations regarding the formulation of restrictions for finding the extremum of the objective Function (15) are identical to those embodied in Restrictions (13) and (14). At the same time, Restriction (13) fully satisfies the statement of the Problem (15), while Restriction (14) can be written in terms of the definition of QνSM:(16)$$y=F\left(\left(X+\nu^{-}+WL_{\nu }\right),\left(\alpha^{-}+PL_{\alpha }\right)\right)+\upsilon^{-}+QL_{\upsilon }.$$

Let us pay attention to the situation when the measurement errors υ(t) and the values of the vector of the initial parameters of the stochastic model s(t) are characterized by non-linearity of the rth degree:$$s\left(t\right)=\sum_{r=1}^{R}\sum_{i=1}^{n}\alpha_{i}^{h}x^{\left(h_{i}\right)\left(t\right)}+\upsilon \left(t\right),$$
where $\alpha =\left(\alpha_{i}\right)$ is a vector of parameters, the independent stochastic elements of which take values from the ranges $A_{i}=\left[\alpha_{i}^{-},\alpha_{i}^{+}\right]$ with the probability distribution densities $p_{i}\left(\alpha_{i}\right)$, $i=\bar{1,n}$.

The measurement of the components of the entities “input” and “output” of the investigated process takes place at moments $t_{j}$, $j=\bar{1,o}$. The entity “input” is represented by a set of r-matrices, $r=\bar{1,R}$, of the form
$$X^{\left(r\right)}=\left(\begin{array}{ccc} x_{1}^{\left(r\right)}\left(t_{1}\right) & \ldots & x_{n}^{\left(r\right)}\left(t_{1}\right)\\ \vdots & \ddots & \vdots \\ x_{1}^{\left(r\right)}\left(t_{o}\right) & \ldots & x_{n}^{\left(r\right)}\left(t_{o}\right) \end{array}\right)=\left(\begin{array}{ccc} x_{1,1}^{\left(r\right)} & \ldots & x_{1,n}^{\left(r\right)}\\ \vdots & \ddots & \vdots \\ x_{o,1}^{\left(r\right)} & \ldots & x_{o,n}^{\left(r\right)} \end{array}\right)$$
and the entity “output” is represented by stochastic elements of the vector $s=\left(s\left(t_{j}\right)\right)$, $j=\bar{1,o}$.

Denoting $\alpha^{\left(r\right)}=\left(\alpha_{i}^{\left(r\right)}\right)$, $i=\bar{1,n}$; $\upsilon =\left(\upsilon \left(t_{j}\right)\right)=\left(\upsilon_{j}\right)$, $j=\bar{1,o}$, we present the Expression (17) in the form
$$s=\sum_{r=1}^{R}X^{\left(r\right)}\alpha^{\left(r\right)}+\upsilon ,$$
where the independent elements of the vector of the variability of measurements of the entity “output” υ take values in intervals ${ϒ}_{j}=\left[\upsilon_{j}^{-},\upsilon_{j}^{+}\right]$ with the probability density functions $Q\left(\upsilon \right)=\left(q_{j}\left(\upsilon_{j}\right)\right)$, $j=\bar{1,o}$.

Let us identify and investigate the variable entropy estimate of the probability density functions $P\left(\alpha \right)=\left(p_{i}\left(\alpha_{i}\right)\right)$, $i=\bar{1,n}$, and $Q\left(\upsilon \right)=\left(q_{j}\left(\upsilon_{j}\right)\right)$, $j=\bar{1,o}$.

We present the objective function of the optimization Problem (12)–(14) in the form
$$H\left(\bar{s}\right)=-\sum_{i=1}^{n}\int_{\alpha_{i}\in A_{i}}p_{i}\left(\alpha_{i}\right)\ln p_{i}\left(\alpha_{i}\right)d\alpha_{i}-\sum_{j=1}^{o}\int_{\alpha_{i}\in A_{i}}q_{j}\left(\upsilon_{j}\right)\ln q_{j}\left(\upsilon_{j}\right)d\upsilon_{j}\to \max.$$

We present the system of Restrictions (13) and (14) in the form
$${\bar{P}}_{i}\left(p_{i}\left(\alpha_{i}\right)\right)=1-\int_{\alpha_{i}\in A_{i}}p_{i}\left(\alpha_{i}\right)d\alpha_{i}=0,\;{\bar{Q}}_{j}\left(q_{j}\left(\upsilon_{j}\right)\right)=1-\int_{\upsilon_{l}\in {ϒ}_{j}}q_{j}\left(\upsilon_{j}\right)d\upsilon_{j}=0,\;\Phi_{j}\left(P\left(\alpha \right),Q\left(\upsilon \right)\right)=-y_{j}+\sum_{r=1}^{R}\sum_{i-1}^{n}x_{ji}^{\left(r\right)}\int_{\alpha_{i}\in A_{i}}\alpha_{i}^{r}p_{i}\left(\alpha_{i}\right)d\alpha_{i}+\int_{\upsilon \in {ϒ}_{j}}\upsilon_{j}q_{j}\left(\upsilon_{j}\right)d\upsilon_{j},$$
where $i=\bar{1,n}$, $j=\bar{1,o}$.

Based on the necessary conditions of stationarity of the Lagrange functional [6,7,8], we will assert that the entropy estimates $E_{\bar{s}}^{\left(1\right)}$ of the probability density functions P(α) and Q(υ) belong to continuously differentiable functions, respectively:(17)$$p_{i}^{\cup }∼a_{i}\exp\left(-\sum_{r=1}^{R}b_{ir}\alpha_{i}^{h}\right),$$
(18)$$q_{j}^{\cup }\left(\upsilon_{j}\right)∼c_{j}\exp\left(-d_{j}\upsilon_{j}\right),$$
where $a_{i}$, $b_{i}$, $c_{j}$, $d_{j}$ are fixed coefficients, $i=\bar{1,n}$, $j=\bar{1,o}$.

The conclusion generalized by Expressions (17) and (18) can be interpreted as follows:

(1) For a linear stochastic model of estimation of variable small data: entropy estimates $E_{\bar{s}}^{\left(1\right)}$ are always exponential functions. The results of measuring the entities “input” and “output” of the investigated process determine the form, and not the type, of the $E_{\bar{s}}^{\left(1\right)}$-functions of the corresponding linear stochastic model;

(2) For a non-linear stochastic model for evaluating variable small data: the nomenclature of the types of functions of entropy estimates $E_{\bar{s}}^{\left(1\right)}$ of the “input” and “output” entities of the investigated process is wider and includes both exponential and power types. The type of $E_{\bar{s}}^{\left(1\right)}$-functions depends on the organization of the measurement process of these “input” and “output” entities.

Therefore, it remains to formalize the variable entropy estimates $E_{\bar{s}}^{\left(1\right)}\left(E_{\tilde{s}}^{\left(1\right)}\right)$ of the probability density functions p and q of the parameters of GνSM(QνSM), respectively. Let us investigate the linear GνSM without taking into account the variability of the measurement of the “input” entity:(19)$$\bar{s}=XpL_{\alpha }+qL_{\upsilon }+\Xi \left(\alpha^{-},\upsilon^{-}\right),$$
where $\Xi \left(\alpha^{-},\upsilon^{-}\right)=X\alpha^{-}+\upsilon^{-}$. We define the a priori probabilities by the elements of the tuple $\langle p^{0},q^{0}\rangle$.

Let us present the objective function of the optimization Problem (15) and (16) in the form
(20)$$H\left(\bar{s}\right)=-\sum_{i=1}^{n}p_{i}\ln\frac{p_{i}}{p_{i}^{0}}-\sum_{j=1}^{o}q_{j}\ln\frac{q_{j}}{q_{j}^{0}}\to \max,$$
and the system of restrictions we present in the form
(21)$$\sum_{i=1}^{n}x_{ji}p_{i}L_{\alpha }^{i}+q_{j}L_{\upsilon }^{j}+\Xi_{j},\;\forall j=\bar{1,o},$$
at $\sum_{i=1}^{1}p_{i}=1$, $\sum_{j=1}^{o}q_{j}=1$.

In terms of the Lagrange function, we present the solution of the mathematical programming Problem (20) and (21) as
(22)$$L\left(\bar{s}\right)=H\left(\bar{s}\right)+\beta \left(1-\sum_{i=1}^{n}p_{i}\right)+\mu \left(1-\sum_{j=1}^{o}q_{j}\right)+\sum_{j=1}^{o}\psi_{j}\left(y_{j}-\sum_{i=1}^{n}x_{ji}p_{i}L_{\alpha }^{i}-q_{j}L_{\upsilon }^{j}-\Xi_{j}\right),$$
where β μ are fixed coefficients and $\psi =\left(\psi_{1},\ldots ,\psi_{o}\right)$ is a set of Lagrange multipliers.

Entropy estimates $E_{\bar{s}}^{\left(1\right)}=\left[\left(p_{i}^{\cup }\left(\psi \right),i=\bar{1,n}\right),\left(q_{j}^{\cup }\left(\psi \right),j=\bar{1,o}\right)\right]$ are determined based on Expression (22):(23)$$0\leq p_{i}^{\cup }\left(\psi \right)=\frac{p_{i}^{0}\exp\left(-\sum_{j=1}^{o}x_{ji}\psi_{j}L_{\alpha }^{i}\right)}{\sum_{i=1}^{n}p_{i}^{0}\exp\left(-\sum_{j=1}^{o}x_{ji}\psi_{j}L_{\alpha }^{i}\right)}\leq 1,\;0\leq q_{j}^{\cup }\left(\psi \right)=\frac{q_{j}^{0}\exp\left(-\psi_{j}L_{\upsilon }^{j}\right)}{\sum_{j=1}^{o}q_{j}^{0}\exp\left(-\psi_{j}L_{\upsilon }^{j}\right)}\leq 1,\;\Phi_{j}\left(\psi \right)=\frac{1}{y_{j}-\Xi_{j}}\sum_{i=1}^{n}x_{ji}p_{i}^{\cup }\left(\psi \right)L_{\alpha }^{i}+q_{j}^{\cup }\left(\psi \right)L_{\upsilon }^{j}=1.$$

Now let us investigate how the formulation and solution of the optimization Problem (20) and (21) will change if interval restrictions $0\leq p_{i}\leq 1$, $\forall i\in \left[\bar{1,n}\right]$, $0\leq q_{j}\leq 1$, $\forall j\in \left[\bar{1,o}\right]$, are respectively imposed on the values of the elements of the stochastic vectors 〈p,q〉.

Under such conditions, the variable entropy estimate $E_{\tilde{s}}^{\left(1\right)}$ of the probability density functions of the parameters p and q of QνSM can be obtained by solving the problem of finding the extreme generalized entropy of the form
(24)$$H\left(\tilde{s}\right)=-\sum_{i=1}^{n}\left(p_{i}\ln\frac{p_{i}}{{\overset{⌢}{p}}_{i}^{0}}+\left(1-p_{i}\right)\ln\left(1-p_{i}\right)\right)-\sum_{j=1}^{o}\left(q_{j}\ln\frac{q_{j}}{{\overset{⌢}{q}}_{j}^{0}}+\left(1-q_{j}\right)\ln\left(1-q_{j}\right)\right)\to \max,$$
where ${\overset{⌢}{p}}_{i}^{0}=p_{i}^{0}/\left(1-p_{i}^{0}\right)$, ${\overset{⌢}{q}}_{j}^{0}=q_{j}^{0}/\left(1-q_{j}^{0}\right)$, $i=\bar{1,n}$, $j=\bar{1,o}$.

The objective Function (24) is supplemented by the adapted balance Equation (21):(25)$$\sum_{i=1}^{n}x_{ji}p_{i}L_{\alpha }^{i}+q_{j}L_{\upsilon }^{j}+\Xi_{j}=y_{j},$$
where $0\leq p_{i}\leq 1$, $0\leq q_{j}\leq 1$, $i=\bar{1,n}$, $j=\bar{1,o}$.

Applying the method of Lagrange multipliers [7,8,36], the extreme entropy estimates $E_{\tilde{s}}^{\left(1\right)}$ for the optimization Problem (24) and (25) will be obtained as a result of solving the system of equations
(26)$$0\leq p_{i}^{\cup }\left(\psi \right)=p_{i}^{0}/\left(p_{i}^{0}+\left(1-p_{i}^{0}\right)\exp\sum_{j=1}^{o}x_{ji}\psi_{j}L_{\alpha }^{i}\right)\leq 1,\;0\leq q_{j}^{\cup }\left(\psi \right)=q_{j}^{0}/\left(q_{j}^{0}+\left(1-q_{j}^{0}\right)\exp\left(-\psi_{j}L_{\upsilon }^{j}\right)\right)\leq 1,\;\Phi_{j}\left(\psi \right)=\frac{1}{y_{j}-\Xi_{j}}\sum_{i=1}^{n}x_{ji}p_{i}^{\cup }\left(\psi \right)\psi_{j}+q_{j}^{\cup }\left(\psi \right)L_{\upsilon }^{j}=1,$$
where $i=\bar{1,n}$, $j=\bar{1,o}$.

The starting point for calculating the variable entropy estimate $E_{\tilde{s}}^{\left(1\right)}$ of the probability density functions of the parameters p and q of QνSM, both in the Interpretation (20) and (21), and in the Interpretation (24) and (25), is the calculation of the Lagrange multipliers ψ as a result of solving the systems of equations represented by Expressions (23) and (26), respectively. This process can be arranged, for example, according to the multiplicative algorithm [36]:$$\varphi_{j}^{k+1}=\varphi_{j}^{k}\Phi_{j}\left(\varphi^{k}\right),$$
where $\varphi_{i}=\exp\left(-\psi_{j}\right)$ are exponential Lagrange multipliers, $\varphi_{j}^{0}>0$, $j=\bar{1,o}$.

## 3. Experiments

Let us demonstrate the functionality of the mathematical apparatus proposed in Section 2 using the example of calculating the variable entropy estimate of the probability density functions of the characteristic parameters of the linear stochastic small data estimation model with the dimension of the entities “input” × “output” of [5]×[2]. The matrix of the measurements of the “input” entity looks like this:$$X=\left(\begin{array}{ccccc} 1.805 & 2.103 & 3.310 & 2.007 & 1.505\\ 4.992 & 3.800 & 2.996 & 2.812 & 1.899 \end{array}\right).$$

The vector of the measurements of the “output” entity, taking into account variability, looks like this:$$y=\left(\begin{array}{cc} 21.091 & 32.814 \end{array}\right).$$

Quasi-moments of the first order are described by the expressions:$$a_{i}=3.333p_{i},\;\alpha_{i}\in A_{i},\;\forall A_{i}\in A=\left[0,10\right],\;i=\bar{1,5};\;\upsilon_{1}=-1+2q_{1},\;\upsilon_{2}=-2+4q_{2},\;\upsilon_{1}\in {ϒ}_{1}=\left[-3,3\right],\;\upsilon_{2}\in {ϒ}_{2}=\left[-6,6\right].$$

The fixed parameters of the reference model are described by the vector
$$\alpha^{0}=\left(\begin{array}{ccccc} 1.011 & 2.212 & 1.918 & 3.986 & 0.996 \end{array}\right).$$

The deviations from the values specified in the vector $\alpha^{0}$ caused by the variability of the measurements are characterized by an error $\varepsilon =\left\|\alpha^{0}-\alpha \right\|/\left(\left\|\alpha^{0}\right\|+\left\|\alpha \right\|\right)$.

Summarizing the given initial information in the format of Expression (19), we obtain:$$XL_{\alpha }p+L_{\upsilon }q=\vec{1},$$
where $L_{\upsilon }=\left(\begin{array}{cc} 0.249 & 0\\ 0 & 0.312 \end{array}\right)$, $XL_{\alpha }=\left(\begin{array}{ccccc} 0.747 & 0.873 & 1.366 & 0.834 & 0.622\\ 1.065 & 0.982 & 0.767 & 0.721 & 0.449 \end{array}\right)$, $\vec{1}=\left(\begin{array}{cc} 1 & 1 \end{array}\right)$.

A priori information about the initial values of the vectors $p^{0}=\left(p_{i}^{0}\right)$, $i=\bar{1,5}$, and $q^{0}=\left(q_{j}^{0}\right)$, $j=\bar{1,2}$, is summarized in the corresponding named sets: $p_{A}^{0}=\left\{1;1;1;1;1\right\}$, $p_{B}^{0}=\left\{0.1;0.2;0.3;0.3;0.1\right\}$, $p_{C}^{0}=\left\{0.3;0.4;0.1;0.05;0.15\right\}$, $q_{D}^{0}=\left\{0.2;0.8\right\}$, $q_{E}^{0}=\left\{1;1\right\}$.

The tuple $\langle p_{A}^{0},q_{E}^{0}\rangle$ implies a uniform distribution of the characteristic parameters p and disturbing influences causing measurement variability, q, respectively. Tuples $\langle p_{B}^{0},q_{D}^{0}\rangle$ and $\langle p_{C}^{0},q_{E}^{0}\rangle$ imply uneven distributions of the characteristic parameters and influences, while the latter represents the variant combined according to the a priori probabilities of the corresponding entities.

We obtain optimization problem Statements (20) and (24) for the initial parameters presented above.

The formulation of the optimization Problem (20) and (21) for the above-mentioned initial data has the form:(27)$$H\left(\tilde{s}\right)=-\sum_{i=1}^{5}p_{i}\ln\frac{p_{i}}{p_{i}^{0}}-\sum_{j=1}^{2}q_{j}\ln\frac{q_{j}}{q_{j}^{0}}\to \max,\;0.747p_{1}+0.873p_{2}+1.366p_{3}+0.834p_{4}+0.622p_{5}+0.249q_{1}=1,\;1.065p_{1}+0.982p_{2}+0.767p_{3}+0.721p_{4}+0.449p_{5}+0.312q_{1}=1;\;\sum_{i=1}^{5}p_{i}=1,\;p_{i}>0;\;\sum_{j=1}^{2}q_{j}=1,\;q_{j}>0.$$

The formulation of the optimization Problem (24) and (25) for the above-mentioned initial data has the form:(28)$$H\left(\tilde{s}\right)=-\sum_{i=1}^{5}\left(p_{i}\ln\frac{p_{i}}{{\overset{⌢}{p}}_{i}^{0}}+\left(1-p_{i}\right)\ln\left(1-p_{i}\right)\right)-\sum_{j=1}^{2}\left(q_{j}\ln\frac{q_{j}}{{\overset{⌢}{q}}_{j}^{0}}+\left(1-q_{j}\right)\ln\left(1-q_{j}\right)\right)\to \max,\;0.747p_{1}+0.873p_{2}+1.366p_{3}+0.834p_{4}+0.622p_{5}+0.249q_{1}=1,\;1.065p_{1}+0.982p_{2}+0.67p_{3}+0.721p_{4}+0.449p_{5}+0.312q_{1}=1;\;{\overset{⌢}{p}}_{i}^{0}=p_{i}^{0}/\left(1-p_{i}^{0}\right),\;{\overset{⌢}{q}}_{j}^{0}=q_{j}^{0}/\left(1-q_{j}^{0}\right),\;i=\bar{1,5},\;j=\bar{1,2};\;0\leq p_{i}\leq 1,\;0\leq q_{j}\leq 1,\;i=\bar{1,5},\;j=\bar{1,2}.$$

Such optimization problems can be solved by methods of non-linear mathematical programming [36]. In particular, for the above optimization problems, the extremum point is analytically identified as $\left(p_{i}^{*}=0.36p_{i}^{0},q_{j}^{*}=0.36q_{j}^{0}\right)$, $i=\bar{1,5}$, $j=\bar{1,2}$. So, for our example, the entropy $H\left(\tilde{s}\right)$ reaches its maximum at the point $\left(p^{*},q^{*}\right)$, where $p^{*}=f\left(i,p_{i}^{0}\right)$, $q^{*}=f\left(j,q_{j}^{0}\right)$, $i=\bar{1,5}$, $j=\bar{1,2}$.

Let us examine these dependencies, taking into account that we previously defined schemes for a priori values: $p^{0}=\left\{p_{A}^{0},p_{B}^{0},p_{C}^{0}\right\}$, $q^{0}=\left\{q_{D}^{0},q_{E}^{0}\right\}$. For clarity, we present the dependences $p^{*}=f\left(i,p_{\left\{A,B,C\right\}}^{0}\right)$ and $q^{*}=f\left(j,q_{\left\{D,E\right\}}^{0}\right)$ in the form of diagrams (Figure 1 and Figure 2, respectively).

More detailed information on the values of the characteristic parameters of the investigated linear stochastic model of the small data evaluation presented in Section 3 can be seen in Figure 3 and Figure 4 (for GνSM and for QνSM, respectively).

These figures visualize the values at the extremum point $\left(p^{*},q^{*}\right)$ of $E^{\left(1\right)}$-estimates of such characteristic parameters as $p_{i}^{\cup }$, $i=\bar{1,5}$; $q_{j}^{\cup }$, $j=\bar{1,2}$, and $H^{*}\left(\bar{s}\right)$ (calculated by Expression (20) adapted to form (27)) and $H^{*}\left(\tilde{s}\right)$ (calculated by Expression (24) adapted to form (28)). At the same time, the schemes of the initial values of the vectors $p^{0}=\left(p_{i}^{0}\right)$, $i=\bar{1,5}$, and $q^{0}=\left(q_{j}^{0}\right)$, $j=\bar{1,2}$, are taken into account.

Comparing the symmetrical values visualized in Figure 3 and Figure 4, it can be concluded that the parameter estimates calculated for interval probabilities (i.e., for QνSM) are characterized by a larger value of the conditional maximum entropy than that inhered for GνSM (i.e., for the normalized probabilities). The theoretical justification of this empirical fact is presented in Section 4.

Information about the state of the linear stochastic models, summarized by Expressions (27) and (28), is supplemented by such calculated data as:

(1) the value at the point of extremum $\left(p^{*},q^{*}\right)$ of the quasi-moments of the characteristic parameters of GνSM and QνSM ($\alpha_{i}^{*}$, $i=\bar{1,5}$),

(2) estimates of the variability of the above-mentioned parameters caused by interferences ($\upsilon_{j}^{*}$, $j=\bar{1,2}$),

(3) the errors $\bar{\varepsilon }$ and $\tilde{\varepsilon }$, which characterize the deviation of the measured parameters 〈α,υ〉 from the reference $\langle \alpha^{0},\upsilon^{0}\rangle$ for GνSM and QνSM, respectively.

These data are visualized in Figure 5 and Figure 6.

From the information shown in Figure 5 and Figure 6 (in addition to the information presented in Figure 3 and Figure 4), it can be concluded that the reference parameters and a priori probabilities are correlated. That is, the closer the values in the scheme of a priori probabilities are to the values of the reference parameters, the smaller the value of the error ε. This interpretation, in particular, explains the superiority of the scheme $\left(p_{B},q_{D}\right)$ over the scheme $\left(p_{C},q_{D}\right)$, because ${\tilde{\varepsilon }}_{BD}<{\tilde{\varepsilon }}_{CD}$.

## 4. Discussion

Let us begin the analysis of the results presented in Section 3 of the applied use of the mathematical apparatus proposed in Section 2 with the fact that the estimates of the parameters 〈p,q〉 obtained as a result of solving optimization Problems (27) (derived from Problem (20), (21) and (28)) (derived from Problem (24) and (25)), turn out to be different in terms of the value of the generalized entropy (Expressions (20) and (24), respectively). We will explain this fact on the theoretical basis of the models presented in Section 2.

To simplify the formulations, we will introduce several renovations. Let us redefine entropy H(s) as H(e)=H(s), where e=〈p,q〉. Accordingly, $e_{1}^{*}$ will be the optimal estimate of the parameters $\langle p^{*},q^{*}\rangle$ represented by normalized probabilities ($H\left(\bar{s}\right)$ variant) and $e_{2}^{*}$ will be the optimal estimate of the parameters $\langle p^{*},q^{*}\rangle$ represented by interval probabilities ($H\left(\tilde{s}\right)$ variant). Let us denote $\hat{e}=\arg\max H\left(e\right)$ and define the sets
(29)$$\bar{E}=\left\{e:\langle p,1\rangle ,\langle q,1\rangle \right\}\subset \tilde{E}=\left\{e:0\leq e\leq 1\right\}.$$

Summarizing what has been entered, we formulate the following: if $\hat{e}\in \left(R_{+}^{\left(n+o\right)}\backslash \bar{E}\right)$ then $H\left(e_{1}^{*}\right)<H\left(e_{2}^{*}\right)$. The equality $H\left(e_{1}^{*}\right)=H\left(e_{2}^{*}\right)$ holds when $e_{1}^{*}=\hat{e}$. Let us explain our conclusions. The analysis of the function described by Expression (20) shows that it is a concave function with a single maximum at the point $\hat{e}$. The value of entropy H(e) depends on the distance of a point e from the extreme point $\hat{e}$. In this context, we denote as $\Delta \left(\hat{e},e_{1}^{*}\right)$ the distance between the extreme point $\hat{e}$ and the point $e_{1}^{*}$, the coordinates of which we obtain as a result of solving optimization Problem (20) and (21). Accordingly, the parameter $\Delta \left(\hat{e},e_{2}^{*}\right)$ characterizes the distance between the extreme point $\hat{e}$ and the point $e_{2}^{*}$, the coordinates of which we obtain as a result of solving optimization Problem (24) and (25). Since Function (20) is strictly concave, based on the Relation (29) we can conclude that $\Delta \left(\hat{e},e_{1}^{*}\right)<\Delta \left(\hat{e},e_{2}^{*}\right)$. The equality $\Delta \left(\hat{e},e_{1}^{*}\right)=\Delta \left(\hat{e},e_{2}^{*}\right)$ holds only when $e_{1}^{*}=\hat{e}$. The presented theoretical explanations explain the discrepancy between those presented in Figure 3 and Figure 4 empirical values of $E_{\bar{s}}^{\left(1\right)}\left(H\left(\bar{s}\right)\right)$ and $E_{\tilde{s}}^{\left(1\right)}\left(H\left(\tilde{s}\right)\right)$ for the same schemes $\left(p_{\left\{A,B,C\right\}}^{0},q_{\left\{D,E\right\}}^{0}\right)$. Comparing the symmetrical values visualized in Figure 3 and Figure 4, it can be concluded that parameter estimates calculated for the interval probabilities (i.e., for QνSM) are characterized by a larger value of the conditional maximum entropy estimate than that characteristic of the normalized probabilities of GνSM. Thus, the mathematical apparatus presented in Section 2 was empirically confirmed in Section 3.

In addition, the results of the experiments presented in Section 3 confirmed the conclusion generalized by Expressions (17) and (18) that, for a linear stochastic model of variable small data estimation, entropy estimates $E_{\bar{s}}^{\left(1\right)}$ are always exponential functions. The results of measuring the “input” and “output” entities of the investigated process determine the form, and not the type, of the $E_{\bar{s}}^{\left(1\right)}$-functions of the corresponding linear stochastic model of small data estimation.

The results shown in Figure 3 and Figure 4 showed that a priori information about the initial values of the vectors $p^{0}=\left(p_{i}^{0}\right)$, $i=\bar{1,5}$, and $q^{0}=\left(q_{j}^{0}\right)$, $j=\bar{1,2}$, summarized in the corresponding named sets of $p_{\left\{A,B,C\right\}}^{0}$, $q_{\left\{D,E\right\}}^{0}$, has a significant effect on the $E_{\left\{\bar{s},\tilde{s}\right\}}^{\left(1\right)}\left(p_{i}^{\cup },q_{j}^{\cup },H\left(\left\{\bar{s},\tilde{s}\right\}\right)\right)$ estimates.

In this context, the fact that the author’s mathematical apparatus allows the calculation of the quasi-momentums of the characteristic parameters $\alpha_{i}^{*}$, $i=\bar{1,5}$, of both the GνSM and QνSM, as well as the taking into account of their variability $\upsilon_{j}^{*}$,$j=\bar{1,2}$, caused by the measurement errors, is very relevant. From those visualized in Figure 5 and Figure 6 of the data, it can be seen that the ε deviations from the values indicated in the vector $\alpha^{0}$ caused by the variability of the measurements are most pronounced for the schemes $\left(p_{C}^{0},q_{D}^{0}\right)$ and $\left(p_{C}^{0},q_{E}^{0}\right)$. These schemes are characterized by the fact that the essential parameters of the models are characterized by an uneven distribution (see Figure 1, “C”), and the influence parameters are characterized by both uneven (see Figure 2, “D”) and uniform distributions (see Figure 2, “E”). For both schemes, we obtained: ${\bar{\varepsilon }}_{CD}=0.30$, ${\bar{\varepsilon }}_{CE}=0.36$; ${\tilde{\varepsilon }}_{CD}=0.33$, ${\tilde{\varepsilon }}_{CE}=0.36$. Therefore, for the considered example, the unevenness of the distribution of parameters $p_{i}$, $i=\bar{1,5}$ provided a significant contribution to the high value of errors ε. Reliable a priori information turned out to be very important in the entropy estimation of variable small data.

## 5. Conclusions

The article analytically summarizes the idea of applying the Shannon entropy maximization principle to sets that represent the results of observations of the “input” and “output” entities of the stochastic model for evaluating variable small data. To formalize this idea, a sequential transition from the likelihood function to the likelihood functional and the Shannon entropy functional is analytically described. Shannon’s entropy characterizes the uncertainty caused not only by the probabilistic nature of the parameters of the stochastic data evaluation model but also by influences that distort the results of measurements of the values of these parameters. Accordingly, based on the Shannon entropy, it is possible to determine the best estimates of the values of these parameters for maximally uncertain (per entropy unit) influences that cause measurement variability. This postulate is organically transferred to the statement that the estimates of the probability distribution density of the parameters of the stochastic model of small data obtained as a result of Shannon entropy maximization will also take into account the fact of the variability of the process of their measurements. In the article, this principle is developed into the information technology of the parametric and non-parametric evaluation on the basis of Shannon entropy of small data measured under the influence of interferences.

The article also examines the structural properties of stochastic models for variable data evaluation, the parameters of which were represented by normalized or interval probabilities. At the same time, the inherent non-linearity of these models and errors in measuring the values of the “output” entity was taken into account.

The functionality and adequacy of the created mathematical apparatus are proven based on the empirical results obtained during the investigation of the linear stochastic model of evaluating specific variable small data.

The authors acknowledge that the research presented in the article is formulated in academic form. This circumstance complicates the applied use of the obtained results. At the same time, the developed methodological approach can be useful in various important applications. In particular, it concerns the assessment of software reliability, when the sample of data is usually not large due to the difficulties of reliably assessing them during the testing and operation of the system. In this case, the lack of testing data or information about failures during pilot software operation can be compensated for by analyzing the assumptions that are specific to the software and selecting appropriate models using assumption matrices [37]. Thus, studies that combine the analysis of small data and expert methods are interesting.

Another important application is in safety critical systems, which, due to multi-level reserving, have as a rule a low failure rate and small data about them. On the other hand, it is extremely important for such systems to have accurate or at least interval estimates of indicators with an acceptable range. For that, the described method could be combined with the traditional methods of reliability analysis and risk oriented assessing of safety indicators using formal and semi-formal methods [38].

In this regard, **further research** is proposed to formalize the obtained information technology on a UML basis. This will allow the future work to reach the stage of implementing the profile framework. In addition, it would be very interesting and useful from a practical point of view to combine Big and Small Data analysis to create universal or adaptable framework focusing on the assessment of data quality and their selection according to the quality indicator.

## Figures and Tables

**Figure 1 entropy-25-00184-f001:**
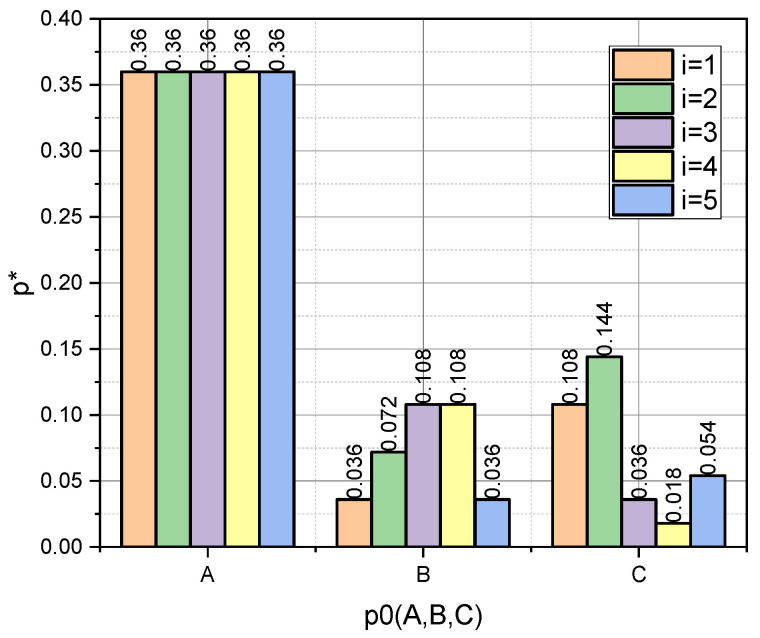
Visualization of dependence $p^{*}=f\left(i,p_{\left\{A,B,C\right\}}^{0}\right)$.

**Figure 2 entropy-25-00184-f002:**
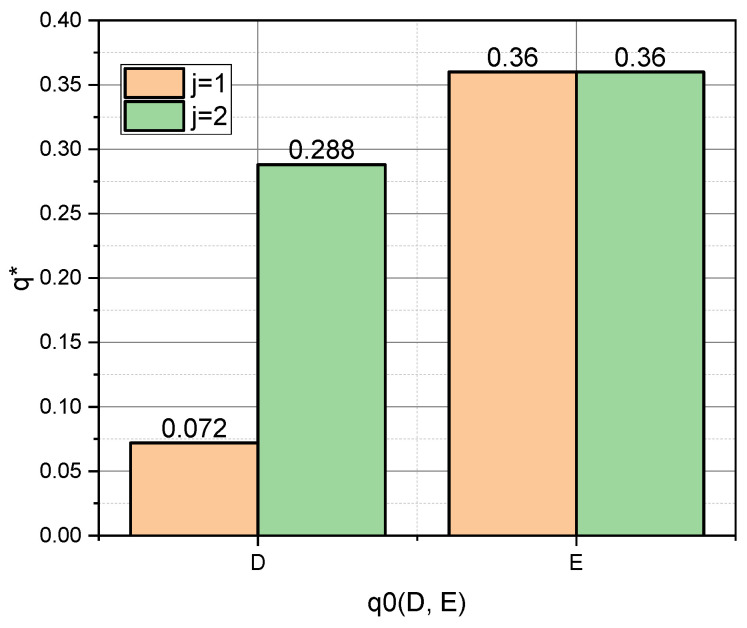
Visualization of dependence $q^{*}=f\left(j,q_{\left\{D,E\right\}}^{0}\right)$.

**Figure 3 entropy-25-00184-f003:**
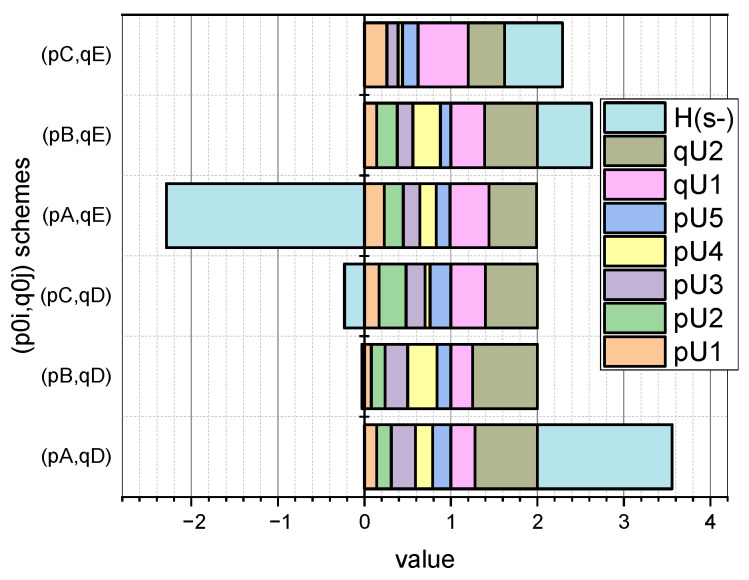
Visualization of dependence $E_{\bar{s}}^{\left(1\right)}\left(p_{i}^{\cup },q_{j}^{\cup },H\left(\bar{s}\right)\right)=f\left(p_{\left\{A,B,C\right\}}^{0},q_{\left\{D,E\right\}}^{0}\right)$, $i=\bar{1,5}$, $j=\bar{1,2}$.

**Figure 4 entropy-25-00184-f004:**
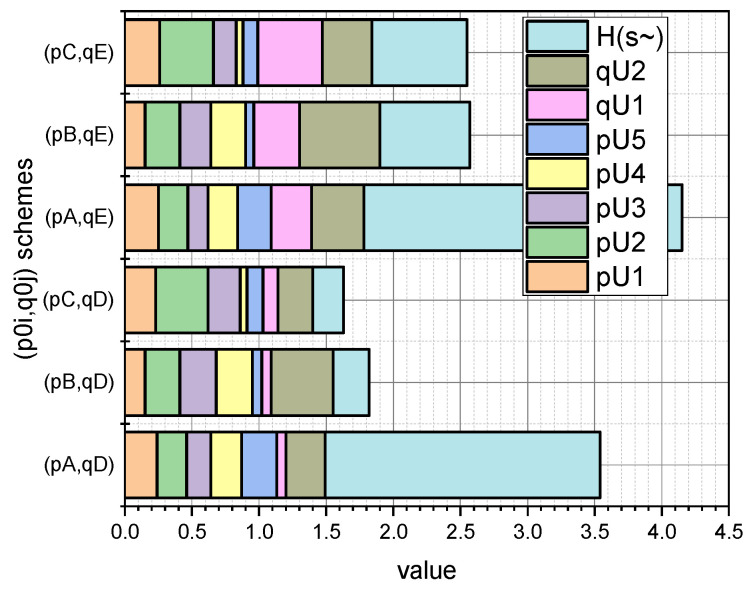
Visualization of dependence $E_{\tilde{s}}^{\left(1\right)}\left(p_{i}^{\cup },q_{j}^{\cup },H\left(\tilde{s}\right)\right)=f\left(p_{\left\{A,B,C\right\}}^{0},q_{\left\{D,E\right\}}^{0}\right)$, $i=\bar{1,5}$, $j=\bar{1,2}$.

**Figure 5 entropy-25-00184-f005:**
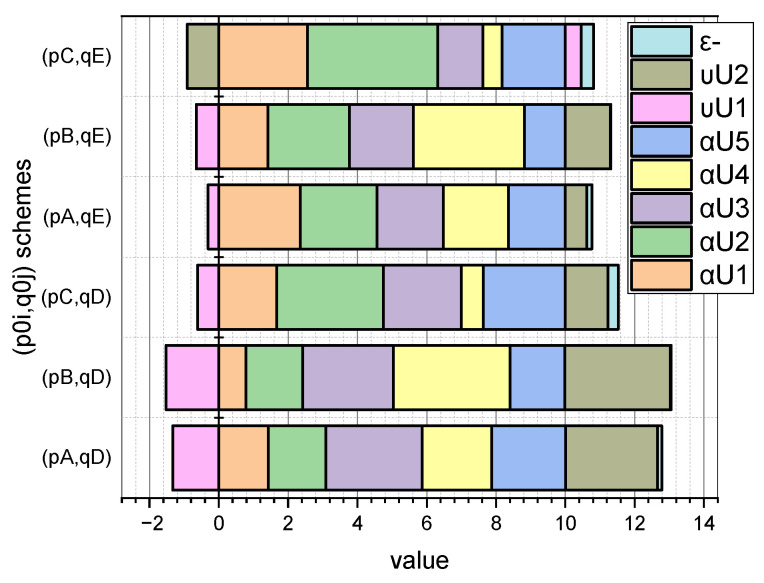
Visualization of dependence $\left(\alpha_{i}^{*},\upsilon_{j}^{*},\bar{\varepsilon }\right)=f\left(p_{\left\{A,B,C\right\}}^{0},q_{\left\{D,E\right\}}^{0}\right)$, $i=\bar{1,5}$, $j=\bar{1,2}$.

**Figure 6 entropy-25-00184-f006:**
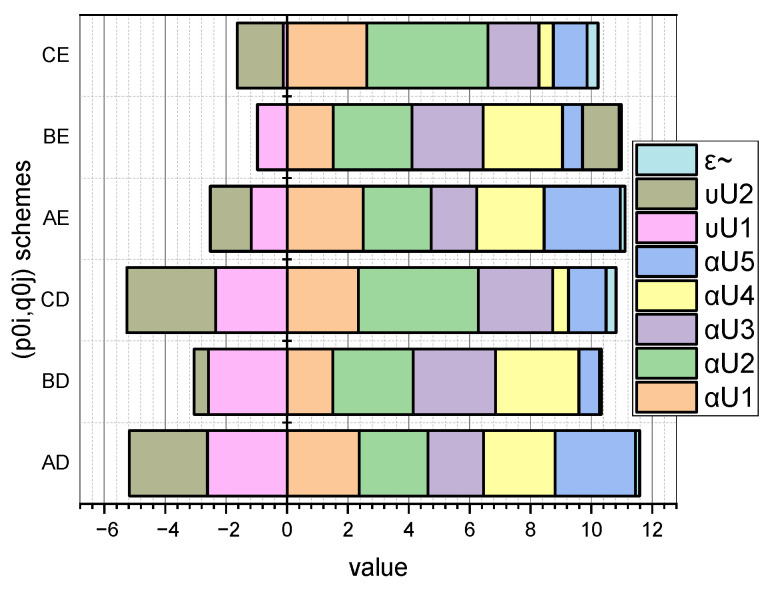
Visualization of dependence $\left(\alpha_{i}^{*},\upsilon_{j}^{*},\tilde{\varepsilon }\right)=f\left(p_{\left\{A,B,C\right\}}^{0},q_{\left\{D,E\right\}}^{0}\right)$, $i=\bar{1,5}$, $j=\bar{1,2}$.

## Data Availability

Most data is contained within the article. All the data available on request due to restrictions, e.g., privacy or ethical.

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
