# Peer review of "Parameterization of the Stochastic Model for Evaluating Variable Small Data in the Shannon Entropy Basis"

_entropy, 2023, doi:10.3390/e25020184_

Round 1

Reviewer 1 Report

This is an interesting article that combines the capabilities of machine learning and mathematical statistics to solve an actual applied problem of small data analysis. The structure of the article is honed. It can be seen that the authors have extensive experience in writing serious scientific papers. The quality of presentation of the material, figures and formulas is sufficient. At the same time, I have a number of recommendations that, in my opinion, can make the article more expressive.

1. The authors should consider the possibility of splitting the Introduction section into the Introduction itself and the Overview of analogues.

2. The article, of course, fits the subject of Entropy journal. However, the authors did not cite any articles from this journal. Hasn't Entropy published strong papers on data analysis?

3. The authors use the mathematical apparatus of Shannon's entropy to build mathematical models. However, entropy is a very flexible concept. Why did the authors not consider the object of study in the Boltzmann or Renyi entropy basis?

4. In the article, the authors proposed several optimization problems. The authors approached the formulation of these mathematical constructions fundamentally. At the same time, the issue of choosing a method for solving these optimization problems was considered in much less detail.

5. An interesting motive is the choice of data, the processing of which was illustrated in the Experiments section. Is it real data or an abstraction? If this is real data, then please provide more data about their source.

Once again, I note that my comments are in the nature of recommendations. This is a strong and thoughtful article. I will support its publication.

Author Response

Reply to Reviewer #1

Dear Colleague,

if January 1 also marks the beginning of the New Year for you, then we sincerely congratulate you on this holiday! In any case, we wish that in 2023 the good you have done will return three times, health will increase, and the well-being trend will grow steadily.

We thank you for the correct, deep and constructive review of our article. Here are our responses to your comments and recommendations:

  1. The authors should consider the possibility of splitting the Introduction section into the Introduction itself and the Overview of analogues.

Dear Colleague, please let us leave Section 1 in its original state. This way we will keep the material of this section connected..

  1. The article, of course, fits the subject of Entropy journal. However, the authors did not cite any articles from this journal. Hasn't Entropy published strong papers on data analysis?

Thank you for bringing our attention to the omission. We have updated the section References in accordance with your comment.

  1. The authors use the mathematical apparatus of Shannon's entropy to build mathematical models. However, entropy is a very flexible concept. Why did the authors not consider the object of study in the Boltzmann or Renyi entropy basis?

Dear Сolleague is a little cunning ) Obviously, the Boltzmann entropy is more suitable for describing physical phenomena, rather than information. At the same time, by mentioning the Renyi entropy, a respected Сolleague "hit the top ten" – this basis will be used by us in further research.

  1. In the article, the authors proposed several optimization problems. The authors approached the formulation of these mathematical constructions fundamentally. At the same time, the issue of choosing a method for solving these optimization problems was considered in much less detail.

Dear Colleague, the adequacy of the mathematical apparatus proposed in the article is ensured by the correctness, consistency and reversibility of the analytical constructions presented in Section 2. Section 3 shows an example to demonstrate the functionality of the proposed approach for small data estimation. This article is the introductory part of a detailed investigation on this topic. In the future, it is planned to devote a separate article to the issue of evaluating the effectiveness of the author's approach in comparison with potential analogues. At the same time, the term "efficiency" will be analytically substantiated in the context of the object of study with access to the appropriate criterion, a detailed comparison will be made and the possibilities of factor analysis to improve the author's approach will be applied. Let us keep the intrigue for now.

  1. An interesting motive is the choice of data, the processing of which was illustrated in the Experiments section. Is it real data or an abstraction? If this is real data, then please provide more data about their source.

Abstract data has been used in this article. However, the formation of real data is already underway. They will be used in the recearch mentioned in p. 4.

Best regards,

Authors.

Reviewer 2 Report

This paper applies the Shannon entropy maximization principle to estimate the probability density of the parameters of the stochastic model of small data. 

The introduction is well-written, and a good overview of the related works is adequate. The contributions are stated in an appropriate way. The numerical results obtained by experiments represent the significance of this approach.

It should be explained in more detail if the necessary conditions are satisfied to write EQs. (2)-(4). Are the corresponding random variables statistically independent? Also, it is necessary to place a reference in some critical parts of the mathematical analysis. If you place a reference in line 200 in front of eq. (2), and in line 210, it would improve the readability of the paper. Is it possible to compare the obtained numerical results with any existing method used to solve this type of problem?

Some minor issues have to be corrected:

·         there is no need to write “probability distribution density function” (e.g. in lines 125, 129, 198, 241…), it is enough to write “probability density function”.

·         it seems that matrix X has dimensions [o × n], not  [o × x]

·         the operator that abbreviates expression i=1,2,…,n has to be defined when appears first time (in line 187)

·         the same variable N is used for the matrix in line 187 and for the interval in line 189 (it can be incorrectly concluded that N_{j,i} represents the {j,i}-th entry of the matrix N)

·         it should be written more strictly what the parameter used in optimization problems, e.g. in maximization in Eq. (15)

·         bullets (or enumeration, if more appropriate) should be used instead of dashes in lines 124-132, 218-223, 240-247, 321-329, 440-447…

·         is it necessary to write the EQs. (10) and (11) in three lines?

Technical preparation of the paper has to be improved:

·         if a sentence is finished with an equation, a dot has to be placed after the equation (e.g. in EQ. (5).

·         there is no need to indent the first line (i.e. to insert a tab) in front of the word 'where' or ‘that’ (e.g. in lines 191, 201, 210, 227, 273, 276,…).

·         please, fix the indentation in EQs. (22), (24), (27), (28)

·         please, check if the double spaces exist in the texts (e.g. in line 441, in 458 before the comma,…

·         some typographic errors have to be corrected (e.g. in line 90: “statics”-> “statistics”)

·         the first references are not written according to the journal template. Please, rewrite it strictly according to the template, in a unified way!

Author Response

Reply to Reviewer #2

Dear Colleague,

if January 1 also marks the beginning of the New Year for you, then we sincerely congratulate you on this holiday! In any case, we wish that in 2023 the good you have done will return three times, health will increase, and the well-being trend will grow steadily.

We thank you for the correct, deep and constructive review of our article. Here are our responses to your comments and recommendations:

  1. It should be explained in more detail if the necessary conditions are satisfied to write EQs. (2)-(4). Are the corresponding random variables statistically independent?

After describing expressions (2)-(4), we added:

Formulating expressions (2)-(4), the authors implied a priori that the measurement results were obtained in accordance with the provisions of the experiment planning theory. The corresponding variables are statistically independent.

  1. It is necessary to place a reference in some critical parts of the mathematical analysis. If you place a reference in line 200 in front of eq. (2), and in line 210, it would improve the readability of the paper.

Thanks for the great idea. We have added relevant footnotes.

  1. Is it possible to compare the obtained numerical results with any existing method used to solve this type of problem?

Dear Colleague, the adequacy of the mathematical apparatus proposed in the article is ensured by the correctness, consistency and reversibility of the analytical constructions presented in Section 2. Section 3 shows an example to demonstrate the functionality of the proposed approach for small data estimation. This article is the introductory part of a detailed investigation on this topic. In the future, it is planned to devote a separate article to the issue of evaluating the effectiveness of the author's approach in comparison with potential analogues. At the same time, the term "efficiency" will be analytically substantiated in the context of the object of study with access to the appropriate criterion, a detailed comparison will be made and the possibilities of factor analysis to improve the author's approach will be applied. Let us keep the intrigue for now.

  1. There is no need to write “probability distribution density function” (e.g. in lines 125, 129, 198, 241…), it is enough to write “probability density function”.

We are grateful for this clarification. The question of the correct mention of terminology in different languages is very important. Corresponding changes have been made to the text of the article.

  1. It seems that matrix X has dimensions [o × n], not [o × x].

You are absolutely right. Corresponding changes have been made to the text of the article.

  1. The operator that abbreviates expression i=1,2,…,n has to be defined when appears first time (in line 187).

Thank you. Corresponding changes have been made to the text of the article.

  1. The same variable N is used for the matrix in line 187 and for the interval in line 189 (it can be incorrectly concluded that N_{j,i} represents the {j,i}-th entry of the matrix N).

Dear Colleague, line 187 gives a description of the matrix N. Line 189 introduces the interpretation of N as an interval matrix.

  1. It should be written more strictly what the parameter used in optimization problems, e.g. in maximization in Eq. (15).

After expression (15) we added the phrase:

Recall that the complex parameter  generalizes a tuple of interval controlled parameters  (see expressions (10) and (5)), and the complex parameter  focuses on the variability of measuring these characteristic parameters (see expression (7)).

  1. Bullets (or enumeration, if more appropriate) should be used instead of dashes in lines 124-132, 218-223, 240-247, 321-329, 440-447…

Thank you. Corresponding changes have been made to the text of the article.

  1. Is it necessary to write the EQs. (10) and (11) in three lines?

No, it's just that we are used to using Elsevier as the initial template - option (2 columns) ) We have made the appropriate corrections.

  1. Technical preparation of the paper has to be improved.

We thank you for these comments. We have tried to find and fix all technical flaws.

To form References, we used the capabilities of the https://citation.crosscite.org/ by selecting the "IEEE" template. We are very sorry if we made a mistake in choosing a valid template. We kindly ask the Technical Editor to advise which of the templates presented on the https://citation.crosscite.org/ is suitable for MDPI.

Best regards,

Authors.

Round 2

Reviewer 2 Report

The paper is improved, but some minor issues still have to be corrected before publication:

 - all variables have to be written in the italic form (e.g. 's' in line 210)

- Eqs (20), (24), and (28) have to be aligned to the right margin, next to the equation numbers

- references should be written according to the Microsoft ford template, which can be downloaded from address:  https://www.mdpi.com/journal/entropy/instructions#preparation.

Author Response

Dear Colleague,

we appreciate your punctuality and patience. Like you, we believe that there are no small things in science.

We have finalized the text of the article and references in accordance with your recommendations.

Best wishes, Authors.
